# Designing on the Basis of Recycling-Metallurgy Possibilities: Material-Specific Rules and Standards for "Anti-Dissipative" Products

**Konrad Schoch** [1,2,*] **, Christa Liedtke** [1,3] **and Katrin Bienge** [1]

1   Research Group Sustainable Production and Consumption, Wuppertal Institute for Climate, Environment and Energy GmbH, P.O. Box 10 04 80, 42004 Wuppertal, Germany; christa.liedtke@wupperinst.org (C.L.); katrin.bienge@wupperinst.org (K.B.)
2   Faculty of Design and Art, University of Wuppertal, Gaußstraße 20, 42119 Wuppertal, Germany
3   Design Faculty, Folkwang University of the Arts, Klemensborn 39, 45239 Essen, Germany
*   Correspondence: konrad.schoch@wupperinst.org; Tel.: +49-202-2492-152

**Abstract:** The demand for metals from the entire periodic table is currently increasing due to the ongoing digitalization. However, their use within electrical and electronic equipment (EEE) poses problems as they cannot be recovered sufficiently in the end-of-life (EoL) phase. In this paper, we address the unleashed dissipation of metals caused by the design of EEE for which no globally established recycling technology exists. We describe the European Union's (EU) plan to strive for a circular economy (CE) as a political response to tackle this challenge. However, there is a lack of feedback from a design perspective. It is still unknown what the implications for products would be if politics were to take the path of a CE at the level of metals. To provide clarification in this respect, a case study for indium is presented and linked to its corresponding recycling-metallurgy of zinc and lead. As a result, a first material-specific rule on the design of so-called "anti-dissipative" products is derived, which actually supports designing EEE with recycling in mind and represents an already achieved CE on the material level. In addition, the design of electrotechnical standardization is being introduced. As a promising tool, it addresses the multi-dimensional problems of recovering metals from urban ores and assists in the challenge of enhancing recycling rates. Extending the focus to other recycling-metallurgy besides zinc and lead in further research would enable the scope for material-specific rules to be widened.

**Keywords:** product design; dissipative loss; EEE; WEEE; urban ores; metals; indium; recycling; circular economy





## 1. Introduction

The ongoing digitalization and rapid diffusion of electrical and electronic equipment (EEE) since the beginning of the 21st century led to an enormous demand for all kinds of metals that are responsible for these devices' functionalities [1,2]. As a result, a massive shift of metal reserves from the ecosphere to the technosphere continues, substantial environmental degradation [3] is being accepted despite the limited availability of metals for humankind [4]. Once there, they become objectified as so-called "designed minerals" [3] by means of ever more complex devices. Generally, the range of built-in functions and the variety of elements used in smartphones and other EEE is increasing [5]. As metals are theoretically infinitely recyclable, the reality differs greatly for a variety of reasons [6–8]. They depart the anthropogenic material cycle [9] and dissipate into the environment without any chance of recovery [10]. As a consequence, they can no longer be used by current and future societies [11]. A driving force causing dissipative losses is an irresponsible design of products, that is, applying new material mixtures not compatible with existing recycling processes [12] as well as building in components that make dismantling and

material separation challenging or even unfeasible [10]. The 30–40% recycling efficiency [9] of the Fairphone 2, probably the most sustainable smartphone [13], is indicative (obviously no tests regarding recyclability of Fairphone 3 have been conducted to date). It must be stated that recycling technologies have not been able to keep pace with the development of advanced and complex products [10]. A trend of growing imbalance between technological (over-) development of EEE and metallurgical processes is becoming more apparent and shows up in the increasing volume of waste of electrical and electronic equipment (WEEE), one of the most rapidly rising waste streams in the world [14]. It has grown by >30% from 2005 to 2020 in the European Union (EU) [15]. Not to mention the fact that an end is far from being in sight as efforts continue to fully implement digital technology in the values of the EU [16,17].

### 1.1. Legal Measures of the EU to Counteract Metal Dissipation

The commitment against increasing dissipation of metals in the end-of-life (EoL) phase of EEE [18] with rising WEEE and for improved recovery of metals from urban ores is partly expressed in the EU's ambitious plan to strive for a circular economy (CE). By means of specific measures, it is supposed to succeed in establishing "a well-functioning internal market for high quality secondary raw materials" [7].

On this path of the EU, the waste hierarchy is of great concern. It is in general meant to protect the environment and human health from the effects of the resulting waste. In the corresponding Directive 2008/98/EC, the member states are called upon to facilitate high-quality recycling of essential materials, such as metals. It argues for appropriate operating mechanisms such as reporting by the member states to the EU on the achievement or non-achievement of specified targets [19].

Additionally, Directive 2002/96/EG on WEEE, which aims at preventing WEEE as a major objective, has been implemented into national law within the countries of the EU [20]. Key statements of amendment 2012/19/EU are an extension in the scope of validity from 2018 on to all EEE and a 5% increase of recycling and recovery rates by 2018 [21]. What is originally intended to be a worthwhile approach simultaneously turns out to be an enormous challenge for recycling technology. It has to face the already mentioned problems on different levels such as fast changes in product design, further miniaturization of components and thin functional layers, large numbers of construction materials, and furthermore the required recovery rate [22]. The producers of EEE are also partly addressed. In accordance with Article 4 of Directive 2012/19/EU, member states should stimulate producers and recyclers to cooperate and take measures to encourage the design and production of EEE, in particular with regard to facilitating the re-use, dismantling, and recovery of WEEE [21,22].

There are other initiatives that aim to improve sustainability issues within the scope of material circularity such as the Ecodesign Directive 2009/125/EC. The underlying idea aims at enhancing the design of EEE in order to achieve better recycling of metals [23]. Parameters are identified that refer to different phases of the entire life cycle of products to set general ecodesign requirements. At the EoL, possibilities for reuse, recycling, and recovery of material should be considered along with Directive 2002/96/EG. Ecodesign parameters for products are, for instance, the number of used materials and components, standard components applied, time needed for dismantling, materials suitable for re-use and recycling as well as easy access to precious components and materials. Implementing ecodesign parameters requires an improved information flow between designers, manufacturers, consumers, and recyclers [24]. The use of critical raw materials and aspects of the CE has been given more attention in preparatory studies carried out in accordance with the methodology for ecodesign of energy-related products (MEErP) [23]. The MEErP is being used to define policy instruments for decreasing the environmental impacts of energy-related products (ErP) within Europe. It was first published in 2005 and is utilized in the framework of Directive 2009/125/EC [25].

However, there are no general requirements that guarantee an increasingly sustainable and closed-loop approach to all EEE placed on the market in the EU [7]. Beyond that, from the perspective of product design, nothing is directly imposed on the manufacturers [25]. One could even say that the willingness to change the design of EEE in order to counteract the dissipation of their inherent metals in the EoL phase is muted by certain legal requirements. An antagonism between rhetorical commitments with a CE in mind and actual legislative efforts appears, for example, in Article 15.5. of the Ecodesign Directive 2009/125/EC. Here it states that measures implemented by it should not have a negative impact on the functionality of EEE from the perspective of the users. In addition, corresponding producers should not be burdened with large administrative efforts [24].

### 1.2. WEEE as a Promising Urban Metal Ore

From the political dimension previously introduced, we take a step forward and look at WEEE such as smartphones as a challenging yet promising urban metal ore. These devices are classified as "small IT and telecommunications equipment" within the scope of the WEEE Directive 2012/19/EU [21]. The average use-time for smartphones in industrialized countries is 2.7 years [26]. It describes the period of time between the first and the last use of an object by the same person, family, or organization and indicates that smartphones are being replaced more often than t-shirts [27]. The time of product exchange is not only being determined by designers and marketers but is reproduced and negotiated constantly in the interaction of all actors in the market [28]. Three levels of relative obsolescence, namely psychological, economical, and technological obsolescence, are known [29]. Some of the main reasons for smartphones being replaced are however limited device functionality, failure to meet the desired requirements, upgrades from the supplier as well as new, more attractive models on the market [27]. In addition to the availability, weight, number, and type of products, their above-mentioned material composition determines the urban mine, which is a source of resource recovery.

Various other factors influence the technological and economic potential to recover metals from these modern mines, especially rapid changes in the design of EEE. As a matter of fact, the possibility of recycling and recovering metals varies as a function of time. Whenever rapidly changing technological innovations are introduced by manufacturers and when products or their components are replaced, recyclers face major challenges in adapting their recycling technology to the new product characteristics. This is clearly evident in the case of display technology, where liquid crystal display (LCD) technology has been replaced by organic light emitting diode (OLED) technology [8,30].

WEEE contains >50 elements in various mixtures and phases [6,8]. From a technological point of view, smartphones have become more functional over time, which has led to an increase in energy demand, storage capacity, and materials required [31]. Even >60 elements are being used to design advanced electronics such as smartphones [32]. Several of them are defined as "critical" by the EU due to supply shortage risks and their economic impact, which is higher than for most other commodities [33]. These include antimony, baryte, beryllium, bismuth, cobalt, gallium, germanium, hafnium, heavy rare earth elements (HREEs), indium, lithium, light rare earth elements (LREEs), niobium, platinum-group elements (PGEs), tantalum, titanium, and tungsten [34]. Although the quantity of each element in a smartphone may seem minor on its own, the combined environmental burden of extracting and processing these valuable metals the phone contains is enormous with 7 billion devices produced globally between 2007–2016 [32].

A closer look at the metal indium, which is used as a semiconductor in displays, reveals the relevance of this. While the content per device is only 0.1 g, the content of all smartphones manufactured since 2007 adds up to 71 tons [31]. This represents the total refinery production of Japan in 2018 or ~10% of the global share in the same year [35]. Other studies such as the project of prospecting secondary raw materials in the urban mine and mining waste (ProSUM) show that the amount of indium used in all screens in European stocks in 2015 is as high as 150 tons [23]. Given this considerable quantity of

metals used, some studies already speak of a considerable raw material treasure in the drawers of German households [36].

WEEE is one of the most rapidly rising waste streams in the world [15] with unpredictable growth potential: the number of devices sold is projected to be 0.6 per capita in 2035, almost twice the amount sold in 2012 (0.35 per capita) [37]. On a global average, up to 18 metals can be currently recovered from urban mines at a rate of >50%. These include titanium, niobium, chromium, manganese, rhenium, iron, cobalt, rhodium, nickel, palladium, platinum, copper, silver, gold, zinc, aluminum, tin, and lead [38].

As one of the largest sink of raw materials, WEEE is of global interest [22] especially regarding critical raw materials and minerals from conflict-affected and high-risk areas [31]. Given this background, the problem of metal dissipation from EEE in their EoL phase, as described above, is not exclusively a problem in environmental terms. Rather, it is a promising challenge that attracts large quantities of valuable metals as well as a lot of money [39]. From this point of view, it is worthwhile to look more energetically for new strategies and concepts such as the CE, which ideally enable the recovery of the raw material treasure from urban ores.

### 1.3. Implications for the Design of EEE

It remains to be seen what the long-term effects of the introduced political measures, which currently have to be implemented mainly by the recyclers, will be. Nevertheless, a look at the rate of circular material use [17] already shows how far removed reality is from the ideal situation promoted by the EU. This was examined in detail [9] using the product-specific approach, which led to the above-mentioned recycling efficiency of Fairphone 2.

Recycling of EEE is an important key for enhancing environmental sustainability, as it not only offers the possibility to prevent the dissipation of metals to a substantial degree but overall has fewer environmental impacts than the primary production of metals [9,17]. However, recycling processes would need to be considered as a basis at the beginning of product design if the path towards a CE at the material level is to be taken by the EU.

Otherwise, the recycling technology would permanently lag behind the rapid technological progress of EEE, resulting in the loss and/or increased use of resource-intensive and critical metals. The described challenge of urgency is therefore framed within the general context of scientific evidence that resource use related to present European production and consumption patterns put at risk the Earth's vital systems and thus society and the very basis of economic development [40]. Indications that mankind is at the edge of its terrestrial possibilities of existence are nothing new, but have been projected for a long while [41,42] and have led to measures assessing environmental impact intensities on the micro level such as the concept of material input per service unit (MIPS) [43,44].

## 2. Building a Material-Specific Knowledge Base

### 2.1. Case Study

The EU's plan to strive for a CE lacks feedback from a design perspective. It is not yet known for which products a CE is already appropriate and for which it is not. Furthermore, there is still a need to provide sufficient guidance on the remaining possibilities of how metals can be used in products. To provide clarification in this respect, we will use this paper to derive initial knowledge for the design of products. For this purpose, the case study of indium tin oxide (ITO), as a relevant application in EEE, is used to link product design directly to the field of recycling-metallurgy and to the assessment of the underlying Metal Wheel's potential [8] as a possible basis for designing products such as EEE.

### 2.2. Methodological Approach

Figure 1 reflects our general methodological approach. We start with the processes of recycling-metallurgy as reflected in the Metal Wheel. From this basis, we derive all possibilities for a CE, that is, the mainly recoverable elements. According to the case study,

this usually concerns the metallurgical infrastructure of zinc and lead, as corresponding carrier metals [8,18,22]. This scope enables us to identify possible functional connections, called material-specific rules. These are considered to be ingredients of "anti-dissipative" products, which would represent an already achieved CE on the material level. Furthermore, the design of electrotechnical standardization is being introduced as a possible tool in order to additionally support the ideally dissipation-free circulation of metals in the EoL phase of products within a CE.

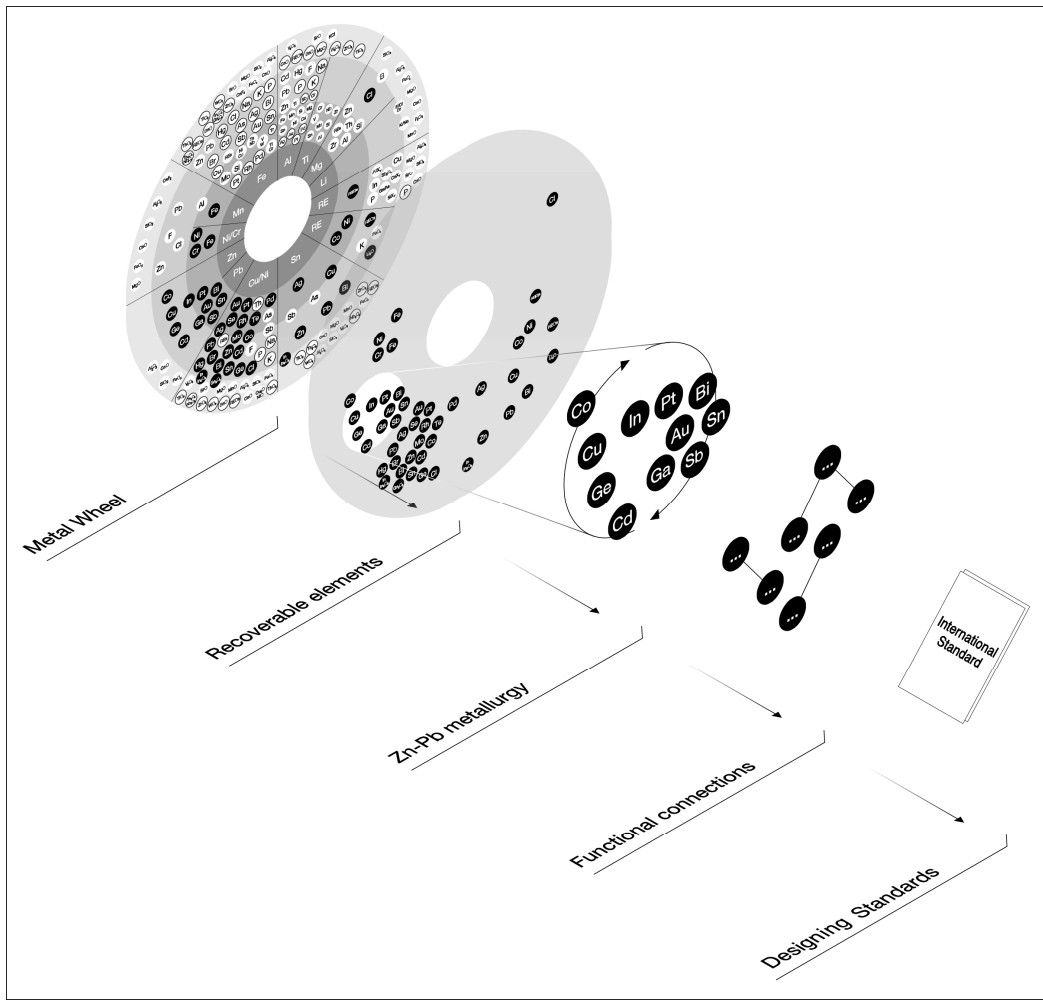

**Figure 1.** Our methodological approach is based on recycling-metallurgical processes as reflected in the Metal Wheel [8]. The further steps are as follows: Deriving mainly recoverable elements; Referring to a specific segment (Zn-Pb metallurgy in our case); Identifying functional connections (material-specific rules within a CE); Providing feedback to product design and politics; Designing standards.

### 2.3. Metal Wheel

As a widely known illustration based on the processes of primary metallurgy, the Metal Wheel maps the recycling physics and shows what actually happens with metals included in EoL devices when they are brought into the processes of their carrier metals. A distinction is hereby made between interlinked hydro- and pyrometallurgy as well as electrorefining/winning infrastructure. Each of the included segments constitutes the entire refinery infrastructure for a carrier base metal refinery [8]. There are a number of different possibilities to link the individual segments of the Metal Wheel in order to recover a larger range of metals. The integrated precious metal copper-lead-nickel metallurgy of Umicore [22,45] as well as the multi metal mining of Aurubis [46] are examples of this. For the purpose

of our case study, we focus solely on the Zn-Pb metallurgy as it is the common and most appropriate system for ITO recycling [47,48].

### 2.4. Electrotechnical Standards

Panels in charge of electrotechnical standardization exist at different levels. In national terms (e.g., in Germany) this is the task of the competence center for electro-technical standardization (DKE). The European committee for electrotechnical standardization (CEN-ELEC) is responsible for the European market in close cooperation with the international organization for standardization (ISO) and the international electrotechnical commission (IEC) for EEE at a global scale [49].

Generally, the potential of standards to actively support a CE has already been recognized [50]. Particular focus is on the level of use and circulation of materials, that is, the aspects of material efficiency [51]. The technical committee 111 (IEC TC 111) globally covers the area of "environmental standardization for electrical and electronic products and systems". In this respect, its aim is to prepare guidelines and horizontal standards to be elaborated with product committees that improve the recycling and EoL management of products in terms of a CE [52].

## 3. Findings

### 3.1. Challenges Associated with ITO

The metal indium has its main application in the production of ITO, a mixed oxide consisting of 90% $In_2O_3$ and 10% $SnO_2$ used in flat panel displays and touch screens of smartphones [53,54]. Taking into account the atomic mass, ITO contains 74% indium. As a cross-sectional technology, flat panel displays are used within different devices. Their sizes vary depending on the device they are used in and range from small scale versions in smartphones to large scale versions in monitors. There are also various technological specifications using ITO as an essential electrode layer [37]. Global refinery production of indium in 2006 was 580 tons [55]. By 2030, its demand is supposed to increase by a factor of ~5.5 [56] to 3300 tons as Figure 2 shows. Indiums share of use as ITO in display technology is predicted to increase from 230 tons in 2006 to 968 tons in 2030 [57] after indium production has already increased by a factor of four from 1994 to 2006 [55].

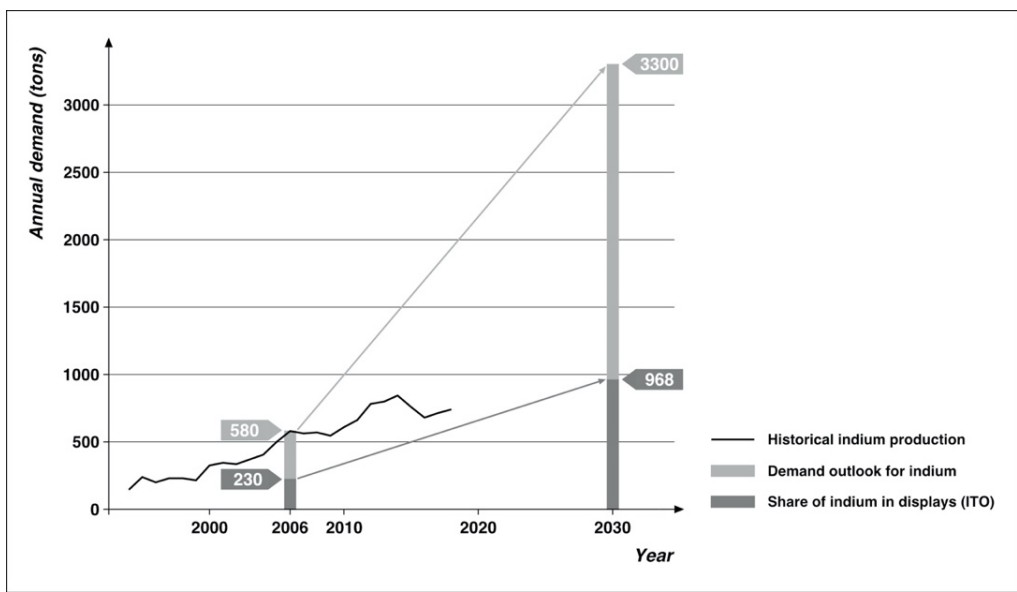

**Figure 2.** Historical production of indium at the global scale from 1994 to 2018 [55]. Demand outlook indicates an increase by a factor of ~5.5 until 2030 (3300 tons) [56] (average of range), compared to global production of indium in 2006 (580 tons). Indiums share in display technology is predicted to increase by a factor of ~4 from 2006 (230 tons) until 2030 (968 tons) [57] (average of range) (own illustration based on [57]).

In low concentrations, indium is mainly contained within polymetallic ores in the earth's continental crust such as sphalerite (ZnS) (up to 0.1%). In smaller quantities, it is also found in other ore minerals such as chalcopyrite ($CuFeS_2$), digenite ($Cu_9S_5$), stannite ($Cu_2FeSnS_4$), and cassiterite ($SnO_2$) [4]. Usually, >95% [58] of indium is recovered as a by-product [59] during zinc production from sphalerite [35]. Therefore, its availability is directly dependent and linked to zinc production [54,60]. Obtaining indium is associated with high environmental burdens as its characterization factor raw material input ($CF_{RMI}$) indicates a total material requirement of 3334 kg/kg, which is five times greater than the amount for lithium and even 21 times greater than that for copper [61].

In the particular case of indium, the main part of the material stream used is currently not recyclable. This is due to the fact that on the one hand recycling is not economically viable as a result of technology and economic barriers, such as a decline in the quality of the materials or low economic benefits compared to costs of recycling [10]. On the other hand, there is the fundamental reason that designed and artificial "minerals" used in urban ores [3] are mixed together in such a complex way that their immanent metals become inseparably connected [8].

Therefore the recycling rate of indium is <1% [38], implying irreversible dissipative losses of indium for current and future generations [11]. The EU agrees on the importance of minerals and metals for the transition of all industrial sectors towards a CE [62]. Mainly for reasons of economic relevance and supply risk, indium has been declared a critical resource by the EU [34]. It should be noted that many other factors influence how the notion of criticality is assessed. Political developments, export restrictions by governments, wars, and civil unrest or sudden increases in demand render the notion of criticality dynamic [63].

A closer look at the metallurgical processes of indium recycling and asking whether or not the use of ITO as a transparent semiconductor in flat panel displays and touch screens of smartphones is justified in terms of a CE seems worthwhile.

### 3.2. Possibilities of Zn-Pb Metallurgy

While the recycling of ITO is not really practiced for the above-described reasons, it is nevertheless possible as demonstrated in the Metal Wheel [8] (and reflected in Figure 1) using the most appropriate metallurgical infrastructure of zinc and lead, as corresponding carrier metals [8,18,22]. The conventional production of primary zinc contains steps for hydrometallurgical removal of indium from leach solution [47], potentially recovering >95% of indium [48]. Therefore, the existing zinc production infrastructure provides access to recover indium from secondary sources. Aimed at ITO recycling, but equally valid for the steps within zinc production, other procedures describe a sulfide precipitation step for the removal of tin out of the leaching solution as well as a zinc cementation step for depositing indium, reaching an even higher indium recovery rate (>99%), which can ensure significant economic value and practicality [64]. All further derived possibilities of the Zn-Pb metallurgy for designing EEE and generally products in terms of a CE on the material level are shown in Figure 3. In total, 11 elements can be obtained that are compatible with the carrier metals zinc and lead and therefore are recoverable in subsequent processing steps. This again results in possible functional connections such as ITO, which are considered to be the scope for designing "anti-dissipative" components and products within a CE.

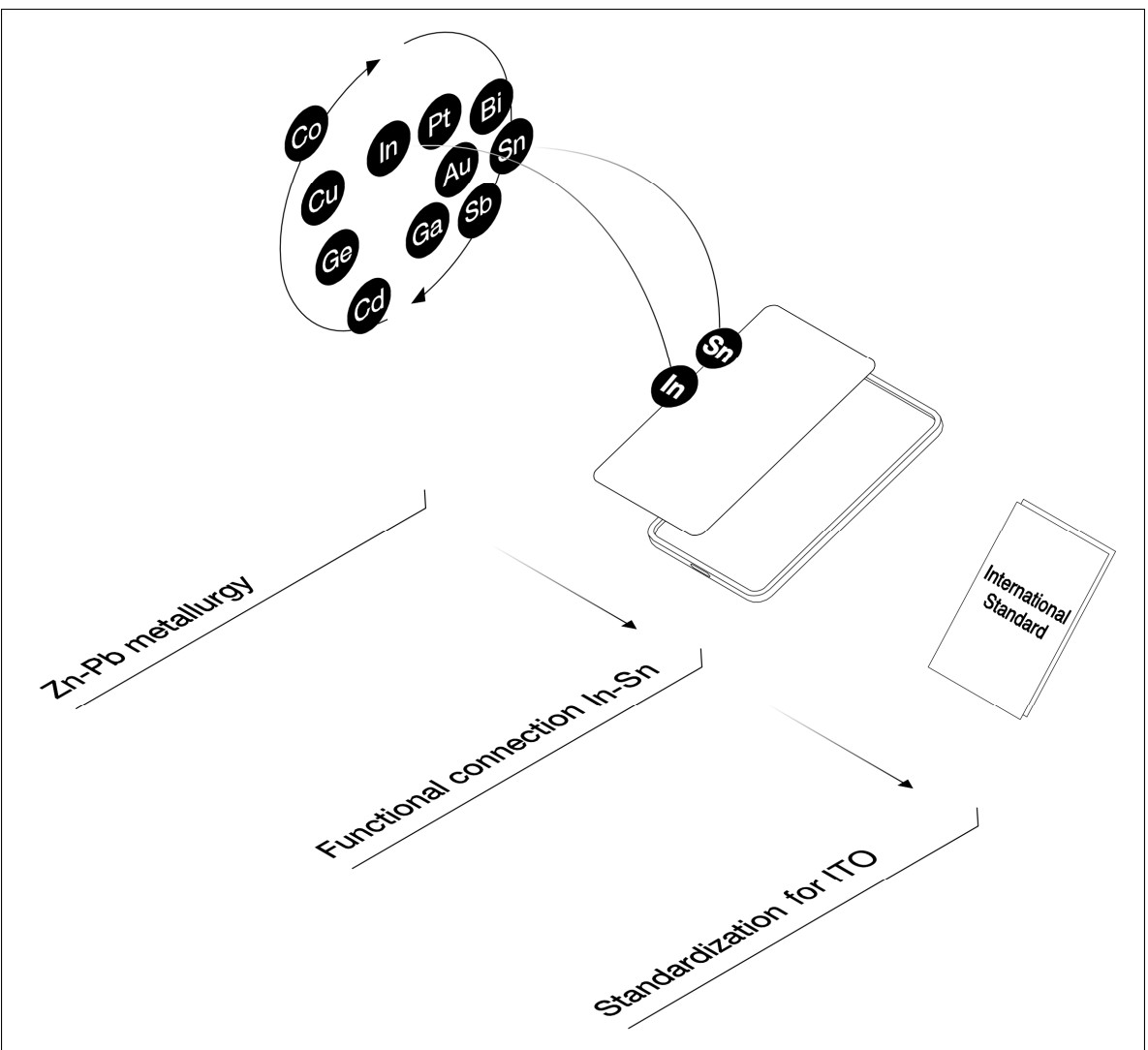

**Figure 3.** Zn-Pb metallurgy [8] possibilities provide material-specific rules on designing "anti-dissipative" functional connections (In-Sn) within EEE such as flat panel displays and touch screens of smartphones. The use of ITO would be appropriate in terms of a CE. Standards at a global scale could support the circulation of indium and tin.

### 3.3. Electrotechnical Standards Improve ITO Recycling

In general, four key actions are at the center of designing a standard. It starts with the submission of an idea with respect to a standardization proposal, which a panel at the national level (e.g., DKE in Germany) decides on accepting. Secondly, the draft standard can be developed together with a group of experts and then submitted to the public for comments. Within a certain period of time (2–4 months), anyone has then the right to submit suggestions or even oppositions to the draft prepared. The national panels such as the DKE then take the drafts forward to the CENELEC and to the IEC. In case there are no substantial modifications or oppositions, the responsible working group can complete the manuscript in a last step and prepare it for inclusion in the corresponding body of standards [65]. Since anyone can submit an application for standardization or even raise objections during its process, it opens up the opportunity for anyone, that is, the general public to participate in political discourses and thus in resource-related issues that would support the recovery of metals from urban ores. As described above, specialized panels such as the IEC TC 111 have been established to promote the implementation of standards within a CE for improved recycling of metals from urban ores [66].

This paper proposes the design and promotion of electrotechnical standards at the global scale for EEE components based on their corresponding and common metallurgical infrastructure, that is, defining a compatible material composition of specific components. Coming from the above-described case study, a standard for flat panel displays and touch screens containing ITO, as is the case for smartphones, would have to be established. At different levels, this could address the current multi-dimensional problems of recovering metals from urban ores such as indium and assist in the challenge of enhancing recycling rates [3] as follows:

(a) international alignment and harmonization with established recycling processes would increase the mass as well as the long-term availability of materially identical components, which could make it more economical to recycle WEEE at all; (b) a major challenge for dismantling centers is to sort components according to their material composition along the right paths in order to enable recycling by type. Materially standardized and moreover clearly and permanently marked components could provide clarity here; (c) if several manufacturers were to use the same components in their products, that is, if different components were reduced to a common level, the price for their production could decrease due to larger order volumes from suppliers. Besides this would also be beneficial in terms of repair prior to recycling, as users would more easily obtain spare parts that are interchangeable between equipment from different manufacturers.

## 4. Discussion

By directly linking the case study to the corresponding recycling-metallurgical processes, the following three important findings could be obtained: (a) the use of certain functional connections within EEE such as ITO are suitable and constitute the basis for their appropriate design in the sense of a CE; (b) vice versa, certain connections are most likely exclusive in principle, meaning that their direct functional connections among metals are not compatible with the available recycling-metallurgical processes. As long as these are in use, recovering desired metals from urban mines will not be possible for current and future generations and thus a CE will remain a distant future; (c) designing standards of appropriate functional connections such as ITO represents a promising tool in order to improve the recycling of inherent metals.

### 4.1. Limitation: The Idea of "Anti-Dissipative" Products

Despite all the enthusiasm for striving towards a CE by the EU [7], which would imply the design of "anti-dissipative" products on the material level, it must be kept in mind that metal dissipation for thermodynamic reasons is quite natural. It occurs not only in the EoL phase but even along the entire life cycle of a product [2,9,18]. The ideal of a CE is therefore not a goal that can really be achieved. Therefore, the focus tends to be on narrowing and minimizing dissipative losses, which certainly is essential to ensure that future generations are not denied access to the urban mines produced today.

### 4.2. Differentiation: With Material-Specific Rules towards a CE

The product-specific approach of Design for Recycling (DfR) is based on the recyclability profile of given products and provides guidelines that support the circulation of their respective ingredients. In essence, it seeks to identify the optimal recycling-metallurgical infrastructure that can ensure the recovery of metals from WEEE [67,68]. In this respect, setting up generalized rules on how to design products with recycling in mind is contradictory due to the increasingly complex and individual nature of each product [3,8].

However, the DfR approach does not provide information on what is possible from the perspective of product design in terms of a CE. If information is to be provided from a design perspective, products would have to be designed and tested from the very beginning on the basis of recycling-metallurgical processes. This implies that designers would have to consider these recycling-metallurgical premises within their design processes. To do so,

they would need material-specific rules. Using this knowledge, they could design and test "anti-dissipative" products from a CE perspective.

*4.3. Conversation: Standards for Resource Management and Regulation*

An electrotechnical standard is proposed in this research that aims at a material harmonization of flat panel displays and touch screens within smartphones that are based on the use of ITO. In this respect, the chosen case study is appropriate as technological development in the field of pixel density of screens has reached such a high level that the human eye is no longer able to recognize individual pixels. Significant further technological enhancement of the resolution is not to be expected as it would not make any difference to the human eye.

In general, highly developed technologies can be strengthened and protected by implementing standards. This enables companies to focus their innovation efforts on other areas. It opens up the possibility for recyclers to plan their recycling processes in the long term and helps political actors to regulate resource management when times of crisis require short-term access to urgently needed resources such as critical metals. It would furthermore counteract the fact that urban ores are not yet reliable sources of raw materials because their material compositions are frequently changing at short intervals due to the rapid pace of technological progress in product design [8].

**5. Conclusions**

Metals and their functional connections such as ITO play a major role in the ongoing digitalization. Additionally, they are crucial for economic strategy [2] and are an essential factor in achieving the sustainable development goals (SDGs) by the United Nations (UN) [69]. As the demand for metals from the entire periodic table is increasing, it is important to understand their recyclable properties. Thus, design decisions can be made that prevent metals from dissipating.

In this paper, we have linked the specific case study of ITO to its corresponding recycling-metallurgical system. This has enabled us to demonstrate, that the use of ITO as a transparent semiconductor within flat panel displays and touch screens of smartphones is indeed possible in terms of a CE. The establishment of standards could furthermore improve the recycling efficiency of inherent metals and the economical motivation for their recycling. This first derived material-specific rule can support the process of product design in terms of a CE. In addition, it can be used to signal politics that it might be appropriate to strive for a CE in this specific case of ITO. Our findings emphasize the importance of further research in this area. Extending the focus to other recycling-metallurgical infrastructures besides zinc and lead would enable the scope for material-specific rules to be widened. This could result in the identification of whole products for which CE ambitions at the material level are already possible today and those for which it is not.

**Author Contributions:** K.S. wrote the paper. He developed the concept, methodological approach and structure. He managed the review process and created the figures. C.L. supported especially in developing the methodological approach. K.B. mainly assisted in setting up the paper's structure. Both supervised the overall process. All authors have read and agreed to the published version of the manuscript.

**Funding:** This research received no external funding.

**Data Availability Statement:** Data is contained within the article.

**Conflicts of Interest:** The authors declare no conflict of interest.

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
