# Peer review of "Designing on the Basis of Recycling-Metallurgy Possibilities: Material-Specific Rules and Standards for “Anti-Dissipative” Products"

_resources, doi:10.3390/resources10010005_

Round 1
Reviewer 1 Report
This paper describes a case study for recycling metals with an example of indium in ITO.
This article itself is interesting for reading, which may attract attention because of recent concern on the environment and the natural resources. However, the reviewer may not think that the contents match the scope for a "research paper" of this journal.
Author Response
Dear reviewer,
attached we send you the cover letter on major revisions regarding your review. Thank you very much for your comments.
best regards
Konrad Schoch

Reviewer 2 Report
Dear Authors,
Your paper “Designing on the Basis of Recycling-Metallurgy Possibilities: Material-Specific Rules and Standards for ‘Anti-Dissipative’ Products” discuss the importance of designing for recycling in order to increase the recyclability. It is well written and easy to read.
Although the topic is interesting, the paper fails in presenting what is new contributions with this research. The concept of design for recycling is known and has been presented in a numerous papers. The following comments could perhaps help in rewriting and improving the paper:
- It is not clear from the introduction where the research front is and how this research will help moving the front further forward.
- It is not clear how exactly the “Metal-Wheel” has been used in the research nor the limitations with the approach.
- A conclusion section can help in highlighting what is new and what the contribution with this paper is
Author Response

(The authors gave the same response as above.)

Reviewer 3 Report
You will find my report in the attached pdf file.

Author Response

(The authors gave the same response as above.)

Round 2
Reviewer 1 Report
The topic dealt with in this paper is interesting for reading but the discussion does not appear to be supported by numerical evidence. The reviewer thinks that a large gap lies between "technically feasible" and "economically sustainable". The authors propose the concept of CE for In as an example but they do not discuss the economic sustainability of it very much.
The reviewer does not think that the contents of this paper meet the requirements for a research paper in an academic journal.
Author Response
Dear Reviewer,
please find attached our cover letter (.docx) for the second round of this review process. Thank you for your comments!
kind regards
Konrad Schoch

Reviewer 2 Report
Dear Authors,
The paper has clearly improved. However, I still have one major concern and that is related to the linkage done between the Metal Wheel and the recycling of ITO. The text on lines 277-282 implies that it would be possible to reach a recovery rate of >99% for indium using Pb or Zn as carrier metal. Looking at the Metal Wheel, Sn would mainly distribute to the carrier metal Pb or Zn and can thereafter be recovered while In will be distributed between the carrier metal, slag, and dust. Reference [60] starts from a used ITO target, which most probably not originates from a Pb or Zn producer, and reaches a very high recovery rate. Please elaborate on how a recovery rate of 99% would be achieved using Pb or Zn as a carrier metal. I agree that both Sn and In are associated with Pb and Zn according to the Metal Wheel and thus can be recovered in subsequent processing steps but not to that level.
Author Response

(The authors gave the same response as above.)

Reviewer 3 Report
You will find my report attached.

Author Response

(The authors gave the same response as above.)

Round 3
Reviewer 1 Report
The reviewer now accepts the authors' claim.
Please improve the resolution of the figures for publication. The characters in the figures are somewhat blurry for reading.
Author Response
Dear reviewer,
Thank you for reviewing our manuscript. Please find attached our response (word document).
Kind regards
Konrad Schoch

Reviewer 2 Report
Dear Authors,
You are discussing an interesting topic in your paper. I agree with you that standards can improve the recycling rate of specific metals and so on. It would be interesting if you in your future work also included the design stage of EE and if that can be improved/structured in similar way so that EE becomes more easily dismantled into its different fractions and therefore also would improve the recycling rate.
Minor comments:
Please have a look at your numbering of references in the text so that it follow the journal guidelines.
Author Response

(The authors gave the same response as above.)

Reviewer 3 Report
Dear Authors, please, see the file attached.

Author Response

(The authors gave the same response as above.)
